# Failure to Thrive in the Outpatient Clinic: A New Insight

**DOI:** 10.3390/nu12082202

**Published:** 2020-07-24

**Authors:** Antonella Lezo, Letizia Baldini, Monica Asteggiano

**Affiliations:** 1Dietetic and Clinical Nutrition Unit, Children’s Hospital Regina Margherita, AOU Città della Salute e della Scienza, 10126 Turin, Italy; alezo@cittadellasalute.to.it; 2Department of Public Health and Pediatrics, School of Pediatrics, University of Turin, 10126 Turin, Italy; monica.asteggiano@unito.it

**Keywords:** failure to thrive (FTT), faltering growth (FG), malnutrition, micronutrient deficiencies (MNDs), outpatients, elimination diet, child neglect, food intake disorders

## Abstract

Failure to thrive (FTT) is an abnormal growth pattern determined by inadequate nutrition. It is a common problem in children, representing 5% to 10% of patients seen in an outpatient setting. Many definitions have been proposed based on anthropometric deterioration; however, they show poor concordance. No single definition is sufficiently sensitive in identifying faltering growth whilst a combination of multiple criteria seems more accurate. In light of the recent redefinition of pediatric malnutrition as a disequilibrium between requirements and intakes of energy, and macro- and micronutrients, a wider conception of FTT as an unsatisfactory nutritional status related to poor growth and health is useful. Although the most severe problems of micronutrient malnutrition are found in developing countries, people of all regions of the world can be affected by micronutrient deficiencies (MNDs), a form of undernutrition with relevant effects on growth and health. Changes in diets and lifestyle, elimination diets, food insecurity, and food intake disorders create the conditions at risk of faltering growth and MNDs. This new insight integrates the “classical” anthropometric criteria for definition and treatment, in the aim of warranting both a regular increase in size and an overall adequate development and health status.

## 1. Introduction

Failure to thrive (FTT) is an abnormal growth pattern determined by inadequate nutrition. Several definitions of FTT are in use, each of them expressing a particular aspect of faltering growth. Diagnosis is generally based on the evaluation of anthropometric parameters, including a sustained decrease in growth velocity [1]. The few studies that have compared different definitions found poor concordance among all the diagnostic criteria for FTT, with no single measure reliably identifying children with significant undernutrition [2].

Recently, it has been argued that FTT is not a diagnosis but rather a physical sign of inadequate nutrition to support growth [3]. The more recent definition “faltering growth” (FG) and malnutrition are frequently associated to express the disequilibrium between nutritional intakes and requirements of macro- and micronutrients [4]. This imbalance hampers overall growth, impacting weight first, then length and head circumference; anthropometric deterioration is the predominant manifestation of FTT, but the development of cognitive skills and appropriate immune function can also be affected, resulting in failure to achieve developmental milestones and good health [5,6]. On the one hand an inadequate nutritional status may be a consequence of several acute or chronic conditions; on the other hand, it leads to an increase in disease burden, mainly due to a decrease in resistance to infections [7].

Contrary to what is reported on anthropometric growth, little is known about the prevalence of nutrient deficiencies in children and toddlers [8]. Even though they require lesser amounts of trace minerals from their diet, they may be more prone to deficiency than adults, because of rapid growth and uneven intake [9,10].

This review offers a wider conception of FTT as an unsatisfactory nutritional status correlated to poor health, looking beyond the “classical” anthropometric criteria for its definition and treatment.

## 2. Classical Definition

Failure to thrive is a broadly used expression that indicates a condition of substandard growth for age and gender in infants and young children. According to the newest indications, we replace the term FTT with the expression “faltering growth” (FG), which is more appropriate and could be perceived as less negative or alarming by caregivers and parents [5,11,12]. Even if FG is a frequently observed condition, there is still no consensus on its definition [12,13].

Commonly used anthropometric criteria are summarized in Table 1 [12,14,15]. Weight for sex and corrected age is the most used parameter, but length/height-for-age and weight-for-length/height are also crucial [16]. It should also be taken into account that some children who falter in growth parameters actually display a normal variant of growth (i.e., catch-down growth in infants born large for gestational age); in these cases, conditional weight, adjusted for regression to the mean, should be considered [17]. Although it is not possible to identify a single threshold that would reliably recognize children who have FG [11], a cut-off around the fifth percentile seems reasonable [12].

Moreover, given the low sensitivity of a single criteria in detecting children with a pathologic growth pattern, a combination of one or more criteria seems more accurate when applied to the general population [12].

To plot individual measurements, the 2006 WHO charts should be used for children up to 2 years of age, while CDC 2000 charts are the reference for children and adolescents aged 2–20 years [4]. For preterm infants, corrected age (number of weeks/months premature + chronological age) should be used until they are 3 years old [17]. Considering that growth is a dynamic process, weight for age or weight for length/height falling in two major percentiles over time is suggested to be considered as a sign of faltering growth [16]. The Z-score of anthropometric variables expresses the individual position in relation to the population reference standards more accurately than percentile cut-offs and is thus recommended for routine use when describing the anthropometric status; many free electronic tools are available, which allow a precise and dynamic evaluation of the patient over time [4,18]. The adoption of a shared and univocal definition and evaluation criteria could make this subject easier to investigate and reduce the risk of under-recognition.

## 3. A New Insight

Being usually due to a caloric intake insufficient to maintain growth, the concept of FG is closely linked to malnutrition. Malnutrition can be defined as an imbalance between macro- and micronutrient requirements and intakes [4]. Furthermore, the diagnosis of malnutrition should not be limited to undernutrition but also include patients with overweight, obesity, and micronutrient deficiencies (MNDs) [19]. MNDs are a form of undernutrition less evident than others and are thus referred to as “hidden hunger” [20]. Coexistence of multiple MNDs frequently occurs and may be coupled with protein or energy malnutrition [19]. Not only do MNDs have a direct impact on individuals and societies, resulting in poorer health, lower educational attainment, and decreased work capacity, but they can also negatively affect immune function, increasing susceptibility to infections [7,21,22,23], since many vitamins and oligoelements are known for contributing to the normal function of the immune system [24].

Children’s growth is a complex process, which comprises several changes, including an increase in the size and complexity of body functions [25]. It is considered to be one of the best indicators of the child’s health and it largely depends on the nutritional status [26]. A deterioration of anthropometric parameters is the most evident sign of FG, but neurocognitive and immunologic development can be affected too. For instance, malnutrition-related changes in intestinal microbiota contribute to growth faltering, dysregulated immune function, and inflammation [27,28,29]. It has been estimated that undernutrition is responsible for more than 45% of all deaths among children under 5 years of age, and the majority of the deaths among undernourished children are attributable to an increased predisposition to severe infections [30]. Moreover, undernutrition has an impact on adult health, educational attainment, work capacity, and earning potential [30,31].

According to this data, a complete definition of FG should take into account both the evident anthropometric deterioration, and the more complex overall nutrition status, which can have either wide-ranging negative health impacts and/or full potential to restore and guarantee good health [19].

## 4. Epidemiology

GF is a common problem in children, representing 5% to 10% of patients seen in an outpatient setting [32,33]. It is estimated that up to 5% of infants and children in the United States have FG [34] but no data exist on the worldwide prevalence of this condition. FG can affect patients of all sexes, races, and ages, but it is most common in infants and younger children. A recent study on children admitted for FG in a tertiary care hospital showed that the majority of them had primary non-organic FG, which is more common in children on public health insurance; this suggests that socioeconomic factors may play a role [33], in accordance with a previous observation that FG is far more common in economically disadvantaged and rural areas [12,35].

As far as malnutrition in developed countries is concerned, it is mostly related to disease. Illness-related malnutrition affects 6–51% of hospitalized children [36,37,38], especially in the presence of chronic diseases. Nevertheless, nutritional support therapy is only given to a small selection of potential candidates, in all probability based on clinical reasons and not on recognition of malnutrition and its severity [39].

MNDs are more frequent than predicted worldwide and outpatients are at higher risk of under-recognition. The World Health Organization (WHO) estimated that more than two billion people suffer from micronutrient deficiency [40]. Iron, iodine, folate, vitamin A, and zinc deficiencies are the most widespread MNDs, and are common contributors of poor growth, intellectual impairment, perinatal complications, and an increased risk of morbidity and mortality [41]. Nearly one-third of the U.S. population is at risk of deficiency of at least one vitamin or has anemia [42]. A global survey of the eicosapentaenoic acid (EPA) and docosahexaenoic acid (DHA) status in blood, from 298 studies, found a “low” or “very low” status (i.e., levels associated with an increased risk of cardiovascular-related mortality) of EPA and DHA in most of the countries assessed [43]. Little is known about the prevalence of nutrient deficiencies for infants and toddlers [9]. Indeed, preschool children and pregnant women have the greatest risk of anemia and zinc deficiency all over the world, and approximately 30% (241 million) of the world’s school-aged children have insufficient iodine intakes [44]. Collectively, the totality of these data strongly suggests that MNDs are prevalent around the globe. Most of these deficits can be reversed by providing missing micronutrients, but—if untreated—can result in irreversible lifelong consequences [19].

Understanding the epidemiology of MNDs is critical for the application of interventional strategies. A Global Database on Child Growth and Malnutrition can be found at this link: https://www.who.int/nutgrowthdb/en/; the Micronutrients database can be found at this link: https://www.who.int/vmnis/database/en/.

## 5. Etiopathogenesis

According to the etiology, FG has been traditionally classified as organic and non-organic. In organic FG, a medical condition is responsible for the imbalance between intakes and requirements, due to three possible mechanisms: Malabsorption, decreased intake, and/or increased metabolism [45]. It is usually resolved with proper treatment according to the diagnosis.

In non-organic FG, psychosocial or family issues are responsible for inadequate intake, so these patients can benefit from behavioral intervention [45,46]. This is the most common scenario in the setting of an outpatient clinic [2]. In a retrospective observational cross-sectional study on 729 children referred to a pediatric gastroenterology outpatient clinic, the most common cause was inadequate nutrition (61.4%), followed by psychiatric and behavioral disorders (17.2%) [47]. This was confirmed by another observational study, led in a pediatric endocrinology outpatient clinic, where nutritional deficiency represented 51.5% of the causes [33]. More than one factor may contribute to poor growth; thus, diagnostic evaluation should not automatically end as soon as one potential cause is identified [48,49]. Some patients can frequently develop a micronutrient deficit, despite adequate caloric intake. As non-organic FG, MNDs are due to multiple, clinical, familiar, and socioeconomic issues, and can result in irreversible lifelong sequelae, depending upon the severity, timing, and extent of the deficiency [50]. Several key nutrients have been described to play a unique role during the first 1000 days of life, including carotenoids, choline, folate, iodine, iron, omega-3 fatty acids, vitamin D, B vitamins, magnesium, and other trace minerals [8]. MNDs often occur as part of an intergenerational cycle. Pregnant mothers without optimal nutritional intakes have children with a suboptimal nutritional status, including impaired physical and mental development [19]. Deficiencies or a suboptimal status in micronutrients can negatively affect immune function and decrease resistance to infections [41], which in turn further impair the nutritional status.

Some groups of patients are particularly prone to developing multiple oligoelements and vitamin deficiency. This review could shed light onto some predisposing conditions for faltering growth and MNDs, which are too often overlooked.

### 5.1. Elimination Diets

#### 5.1.1. Food Allergies

The burden of food allergies (FAs) is growing worldwide, especially in developed countries [51]. Cow’s milk, egg, soybean, wheat, peanut, tree nuts (e.g., almond, cashew, hazelnut, and walnut), fish, and shellfish cause >90% of FA in children [52]. Additionally, parent-perceived FA rates are becoming higher, leading to severe, and unnecessary, exclusion diets [53]. Food allergy is most common in the first few years of life, with an estimated prevalence of nearly 6–8% in childhood [54]. This is a vulnerable period of rapid growth, when inadequate nutrition due to an unbalanced elimination diet could have serious consequences, including MNDs and FG, with an influence on adult health [55,56]. In addition, food aversion, food refusal, food neophobia, and anxiety about eating in general may occur, leading to poor intake [52]. Consequently, a diagnosis of FA should always be confirmed by a specialist followed by close monitoring of growth parameters and nutritional management by an experienced dietitian, in order to provide appropriate alternatives to supply necessary nutrients [56].

Milk and wheat are major constituents of a developing child’s diet. Wheat provides complex carbohydrates, which are the primary source of energy of the brain and should represent 45–65% of a child’s daily energy intake. Besides, wheat contains a unique set of micronutrients (thiamin, niacin, riboflavin, iron, folic acid) that are not present in most fruits and vegetables: When a wheat allergy is present, alternative grains must be provided to fill the nutritional gap [57].

Cow’s milk allergy (CMA) is the most common FA in childhood, whose peak of incidence is reached in the first year of life [54]. Due to its prevalence and early onset, CMA is an important cause of FTT. Dietary elimination of cow’s milk (CM) protein is mandatory, but adequate intake of macro- and micronutrients should be assured [58]. Inappropriate substitutes for milk protein are frequently used [52,55]. Several studies have reported that children affected by CMA are significantly shorter in height [59] and the reintroduction of cow’s milk has been associated with catch-up growth [51,60]. Nearly 50–60% of allergic children have a low intake of vitamin D (<5 μg/day) while only 12% of them are estimated to take supplementation. Moreover, children with CMA are reported to have multiple low intakes: Energy, lipids, proteins, calcium, vitamin D, riboflavin, and niacin. Apart from an insufficient calcium intake, poor growth may be due to the lack of whey proteins, as there is an association between protein-rich diets and growth [61], and animal studies suggest that bone growth is much more resistant to energy deficiency than to protein deficiency [62]. Notably, dairy products seem to play a unique role in catch-up growth, which cannot be substituted by the consumption of meat, fish, and eggs [63]. Therefore, there is a need to prescribe balanced elimination diets, in order to control symptoms, ensure fast nutritional recovery, and avoid malnutrition [64].

#### 5.1.2. Lactose Intolerance

Lactose intolerance (LI) is due to the inability to hydrolyze lactose, due to absent or reduced lactase in the small intestine, which splits the bond linking glucose and galactose [65]. LI can be primary, attributable to a relative or absolute absence of lactase, or secondary due to acute or chronic small-bowel mucosa injury. While congenital lactase deficiency is extremely rare, primary acquired LI is relatively common among children and adolescents after 2–3 years of age [66]. Treatment consists in the use of lactase-treated dairy products or oral lactase supplementation, limitation of lactose-containing foods, or complete dairy elimination.

There is sufficient evidence that dietary lactose enhances calcium absorption and, conversely, that lactose-free diets result in lower calcium absorption [67,68,69]. The avoidance of dairy products to control symptoms has an impact on bone mineralization, although the effects on the risk for osteoporosis later in life need to be clarified [66]. However, calcium absorption is also affected by protein intake, vitamin D status, salt intake, genetic, and other factors that may be the reason for the heterogeneity of previous findings [68,70]. Absorption of other minerals like magnesium and zinc may be reduced by lactose deficiency [68,71].

Patients with varying degrees of lactase deficiency can often tolerate varying amounts of dietary lactose, so low-lactose and lactose-free formulas have no clinical advantages compared with standard lactose-containing formulas for infancy, except for severely malnourished children [72]. Breastfeeding should never be suspended [66]. In conclusion, avoidance of dairy products not usually necessary and is also potentially harmful, and calcium must be provided by alternate non-dairy dietary sources or as a dietary supplement to individuals who avoid milk intake [73].

#### 5.1.3. Vegetarian/Vegan Diet

No large-scale study exists about growth in vegetarian and vegan children. According to a study conducted in Germany, which examined the anthropometric parameters as well as the energy and macronutrient intake of vegetarian and vegan children, compared to omnivorous children, no significant difference was found between groups [74]. On the contrary, other studies reported vegetarian/vegan children younger than 5 years of age to be thinner and smaller than the reference populations [75,76].

A completely plant-based diet is suitable during all stages of life, from pregnancy to lactation and childhood, but it must be well-planned. Due to increasing prevalence, particular attention must be paid, and medical and expert dietetic supervision is recommended to provide advice regarding supplementation and avoid deficiencies [77,78].

In regard to protein needs, they can be easily met by including beans, grains, nuts, seeds, and green leafy vegetables in meals, as they are a good source of protein in vegan diets, although meeting systematically required intakes may be challenging [79]. During infancy and early childhood, breast milk and plant-based formulas can provide a good amount of protein, as homemade plant-based milks cannot ensure the amount of minerals and vitamins required for this age [77].

Excluding animal derivatives, iron is acquired in the non-heme form from plant foods. None of the iron forms are highly absorbed, but non-heme iron has a lesser bioavailability than heme iron (<5% vs. 12–25%) [80]. Consequently, considering also the elevated amount of phytates, even if the iron content of vegan diets is higher than in lacto-ovo-vegetarian or omnivorous diets, the rate of absorption is unpredictable. Some tips like adding vitamin C or other organic acids (e.g., citric acid, malic acid), carotene, and retinol are useful to increase non-heme iron bioavailability [80]. Similarly to iron, zinc absorption is impaired by phytates, fiber, and lignins, and processing grain or other plant foods causes significant nutrient losses [81].

Attention should be paid to eating adequate amounts of calcium-rich foods (e.g., green leafy vegetables low in oxalates, cruciferous vegetables, sesame seeds, almonds) and fortified plant-based milks and plant-based dairy alternatives [77] while vitamin D status depends more on sun exposure and supplementation than on diet, so the recommendations for supplementation are the same as for the general population [82].

Finally, vegan infants are at risk of vitamin B12 deficiency as plant foods lack it. The consumption of B12-fortified foods in vegan diets can be useful, but supplementation is recommended since the beginning of complementary feeding, at around 6 months of age [77,83]. The ESPGHAN Committee on Nutrition recommended that iron, zinc, calcium, vitamin B12, B2, vitamin D, vitamin A, and omega-3 PUFA (DHA) is carefully monitored in different vegetarian and vegan diets because of the high risk of deficiency [78].

### 5.2. Family Dysfunction and Food Insecurity

FG is strongly associated with social and/or familial issues. Firstly, it can be a clue for parental dysfunction, which can be related to several factors, ranging from psychological stresses (i.e., depression, poor self-esteem), to marital strife, psychosis, or work overwhelm [84]. Secondly, FG can be a sign of child neglect and/or abuse [85]. Neglect should always be considered, especially when there is a serious risk of medical complications, worsening medical conditions despite a multidisciplinary approach, or non-compliance with the treatment plan.

Last but not least, FG can be a consequence of food insecurity, a condition defined as the disruption of food intake or eating patterns because of a lack of money or resources [86]. The unavailability of food or money to purchase food accounts for most of the current malnutrition worldwide [49,85]. Food insecurity is associated with all forms of malnutrition, including obesity. Like undernutrition, obesity is associated with a non-various and unbalanced diet, leading to MNDs. Thus, although the most severe consequences of MNDs are found in developing countries, people all over the world can be affected [41]. Recent changes in diet and lifestyles, rising incomes, and increased consumption of convenience foods, together with a reduction in physical activity levels, are possible causes. Nutrition surveys report low levels of fruit and vegetable intake, inadequate intake of important nutrients, and high intake of energy-dense nutrient poor (EDNP) foods in all age groups [85]. Pregnancy and the perinatal period are particularly at risk [85]. Screening for food insecurity should be an integral part of children’s periodic check-ups, based on the understanding of the social and cultural context, in order to help identify families at risk of nutritional deficits [49,61].

### 5.3. ARFID (Avoidant/Restrictive Food Intake Disorder)

Non-organic feeding disorders (NOFEDs) is a term that indicates a condition of deviating feeding behaviors in children, such as food refusal, aversion to feeding, selective eating, and low food intake, without an underlying organic disease [87].

Recently, avoidant/restrictive food intake disorder (ARFID) has been introduced as a new specific diagnosis in the fifth edition of the diagnostic and statistical manual (DSM-5) [88]. It is defined as an eating or feeding disturbance with persistent failure to meet appropriate nutritional and/or energy needs, associated with one (or more) of the following: Significant weight loss (or failure to achieve expected weight gain or faltering growth in children); significant nutritional deficiency, dependence on enteral feeding, or oral nutritional supplements; and marked interference with psycho-social functioning [89]. Interestingly, both FG and nutritional deficiency are part of the definition. If the relationship between NOFEDs and FG is well established, the association with nutritional deficiency in a broad sense is quite a new concept. Restrictive behavior can induce specific deficiencies related to the nature of the excluded foods, even in the absence of faltering growth [90,91,92]. Electrolyte imbalances, fat-soluble and B vitamins deficiency as well as mercury toxicity may be associated problems [93,94,95]. Harshman et al. demonstrated that children with full or sub-threshold ARFID have a significantly higher intake of added sugars, and a lower intake of vitamins K and B12, consistent with limited vegetable and protein intake compared to healthy controls [96].

According to the current evidence, we support the need for diet diversification as part of therapeutic interventions for ARFID to reduce the risk of nutrient insufficiencies and related complications [96].

## 6. Approach

### 6.1. Clinical History

Usually, FG can be defined and classified through a detailed medical history and physical examination [97]. The interview should collect relevant information about family, medical conditions, dietary habits, and socioeconomic issues, which are summarized in Table 2.

As far as bottle-fed infants are concerned, it is important to discuss the formula preparation process and administration to highlight possible mistakes. Furthermore, past medical history should look for any chronic illnesses, hospitalization, recurrent infections, and developmental delay. The latter has both a higher incidence in children with FG [98] or can cause or exacerbate feeding problems.

### 6.2. Intakes Evaluation

To define the deficit, patients’ caloric intake should be compared to requirements. Caloric intake can be calculated thanks to a 24-h food recall or 3-day food diary [99,100], eventually with support of image-assisted approaches [101]. This analysis should preferably be carried out by a trained professional, such as a dietician or a nutritionist, especially for patients where a pathologic condition is suspected [102]. Indirect calorimetry is the most reliable clinical tool to predict the basal energy requirements (BERs) [103], but it is predominantly recommended for research purposes or in critical care. BER can be estimated by using the equations endorsed in the 1985 report of the FAO/WHO/UNU expert consultation [104] and the total energy requirement is calculated as multiples of BER corrected by the physical activity level (PAL): EER = BER × PAL. PAL is defined as the total energy required over 24 h divided by the basal metabolic rate over 24 h [105]. PAL values of 1.4, 1.6, 1.8, and 2.0 approximately reflect low active (sedentary), moderately active, active, and very active lifestyles, respectively [102].

Beside caloric intake, attention should be paid to daily intakes of each macro- and micronutrient, comparing them to recommended intakes as indicated by country-specific guidelines, e.g., Dietary Reference Intake in the U.S. [106] and LARN Livelli di Assunzione di Riferimento di Nutrienti ed energia per la popolazione italiana in Italy [107]. Unlike macronutrients, micronutrient intakes are difficult to assess, and the need for validation studies of tools estimating zinc, selenium, dietary fiber, sugars, and sodium has been highlighted [100].

### 6.3. Physical Examination and Diagnostic Testing

To objectively evaluate FG, as previously explained, accurate anthropometric measurements are crucial. Furthermore, physical examination should look for signs of organic disease and neglect, and of possible MNDs (Table 2). Patients with alarm signs should be hospitalized.

It has been proven that admission to hospital and laboratory testing is unlikely to lead to a specific organic diagnosis in a child whose failure to thrive is unexplained after careful history taking and physical examination [108]; consequently, routine laboratory testing is not recommended, as it has been shown to identify a definitive cause for FG in <1% of children [109,110,111].

Normal variance of growth should be considered as a possible explanation for FG in children with no sign of underlying disease or suggestive medical history (i.e., catch-down growth, familial short stature, constitutional growth delay).

Detailed medical history and physical examination are generally sufficient to start treatment [97,112]. Patients with conditions predisposing to a high risk of MNDs should be tested for micronutrient plasma levels.

## 7. Treatment

With FG being a sign of undernutrition, nutritional repletion is the basis of successful therapy [48] and the treatment’s goal is to establish an optimal growth velocity and guarantee adequate cognitive development [5,45,113]. A multidisciplinary team is usually necessary, as the intervention should focus both on behavioral and nutritional issues. In Figure 1, the classic approach to FG in an outpatient setting is summarized, together with our suggestion for a complementary approach, which should be carried out in parallel.

When undernutrition is caused by psychosocial and familial factors, patients usually benefit from behavioral intervention [45,114], which mainly involves meal regularization (eliminate grazing, scheduled meals, limited duration), an appropriate setting (e.g., highchair, adequate tools, limit distractions), and triggers elimination [46].

Furthermore, age-appropriate nutritional counselling should be provided but is often overlooked. Daniel et al. reported that, out of 97 patients referred to a pediatric endocrinology clinic, 70% had no nutritional assessment or nutritionist consultation prior to referral [33].

For breastfed infants, breastfeeding attachment should be observed and supported, if necessary; for formula-fed infants, proper preparation of formula should be assured. Caregivers of toddlers and older children should be counselled on nutrient-rich healthy food choices, ideally provided in three meals and three snacks per day [112]. Food-Based Dietary Guidelines (FBDGs) have been described as science-based recommendations for different food assumptions in the European population, and can be a useful tool in order to provide advice on foods, food groups, and dietary patterns to caregivers [115].

Families with problems obtaining nutritious foods (food insecurity) should be provided with social work support or other community resources [45]. When caloric requirements are difficult to meet, adding modular or complete oral nutritional supplements is an option [116].

As far as MNDs are concerned, food-based strategies are crucial, namely producing foods that are naturally rich in micronutrients and promoting healthy and diversified diets [41]. Contrary to selective diets, it is worthy to highlight the proven advantages of the Mediterranean diet on health [117]. This diet offers a wide range of components that are proven to be beneficial, like vitamins, polyphenols, good fats (monounsaturated and polyunsaturated fatty acids), and prebiotic fibers, and at the same time, limited amounts of animal origin foods. It has been suggested that foods (as well as viruses, bacteria, physical activity, chemical or psychological stress) to which humans are exposed from intrauterine life to old age activate a series of stress-responding mechanisms that, acting as hormetins, can minimize the effects of inflammatory stimuli, leading to a lifelong adaptation [118,119].

Nonetheless, supplementation does have a role: Not only is it necessary among groups at high risk and in emergencies, but it is also associated with a lower risk of deficiency even in well-nourished individuals [120]. Calder et al. recommend a multivitamin and mineral supplement in addition to the consumption of a well-balanced diet that supplies the basic micronutrient requirements (e.g., RDA) for vitamins and minerals, in specific at-risk conditions, as infections and other stressors can reduce the micronutrient status in the body [7]. Notably, supplementation above the RDA is recommended for vitamins C and D, which promote optimal immune function and may help to control the impact of infections [7]. A recent Cochrane review of randomized clinical trials (80 trials with 205,401 participants) in children 6 months to 12 years of age indicates a positive effect for zinc supplementation in reducing all-cause and infectious disease mortality and a small positive impact on linear growth [121]. Zinc supplementation during pregnancy is associated with a significant reduction in preterm births without an effect on infant birth weight [122]. No routine supplementation recommendations currently exist for the prevention of zinc deficiency.

To summarize, diet habits are the cornerstone of FG prevention and treatment, but supplementation, both universal and individual, can play a key role too.

## 8. Conclusions

FG still remains a widespread problem in pediatrics, but it is often overlooked, especially in the outpatient context. The lack of a uniform definition may be responsible for under-recognition of the prevalence of FG, and have an impact on outcomes in children. Many definitions have been proposed, solely based on anthropometric deterioration. In light of the recent redefinition of pediatric malnutrition, a wider conception of FG as an unsatisfactory nutritional status related to poor growth and health is necessary. Micronutrient deficiencies are more frequent than expected and have relevant effects on growth and health. Even if accurate measurement remains crucial in monitoring a child’s growth, this new insight integrates the “classical” anthropometric criteria in its definition and treatment, aiming to guarantee both a regular increase in size and an overall adequate health status and development.

## Figures and Tables

**Figure 1 nutrients-12-02202-f001:**
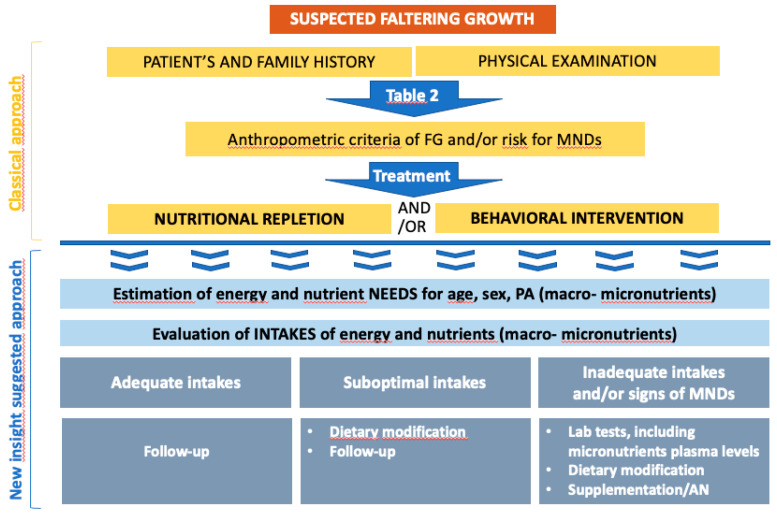
Classical and additional treatment of faltering growth. PA: physical activity; AN: artificial nutrition.

**Table 1 nutrients-12-02202-t001:** Faltering growth: most commonly used anthropometric criteria.

●	BMI for age < 5th percentile
●	Length-for-age < 5th percentile
●	Weight-for-age < 5th percentile
●	Weight < 75 % of median weight for age (Gomez’s criterion) [15]
●	Weight < 80 % of median weight for length (Waterlow criterion) [14]
●	Weight deceleration crossing two major percentile lines

**Table 2 nutrients-12-02202-t002:** Suggested evaluation of a patient with faltering growth (FG). In bold, alarm signs. BMI = Body Mass Index; BP = Blood Pressure; BT = Body Temperature; HR = Heart Rate; RR = Respiratory Rate.

HISTORY	
Family history	Is there a history of Intestinal Bowel Disease? Coeliac disease? Cystic fibrosis? Is there a history of FG? Developmental delay? Abuse and/or neglect? Is there a history of feeding difficulties? How tall are parents? Does he/her have any siblings? Is their growth/development regular?
Background	What is the financial status? The living conditions? Are there any risk factors for neglect/abuse? Poverty, social isolation Single parent (post-partum) depression Are parental coping skills good? Is child-parental interaction good?
Eating habits	Is food insecurity an issue? Does the chil have suck-swallow problems? How is formula prepared? Are there any feeding difficulties? Is the diet restricted/balanced? Food allergies/Intolerance Vegan/Vegetarian diet Selectivity Junk food Is feeding routine regular?
Antenatal and perinatal	Was the pregnancy uneventful? Exposure? Did the mother take supplementation during pregnancy? Was the child born at term? Was his/her body weight low or high? Was/is the child breastfed?
Medical history	Is there a history of recurrent infections? Are there any signs of malabsorption (diarrhea, abnormal stools)? Is the child allergic/atopic? Does he/she present recurrent vomit? Does he/she have reflux?
PHYSICAL EXAMINATION	
Anthropometry (Z-scores)	Weight Length (<2 years) or height BMI or wt-for-lt Cranial Circumference
System review	Are vital signs normal? (Blood pressure, heart rate, body temperature, saturation, respiratory rate) Are there signs of dehydration? Are dysmorphic features present? Systems examination: are there signs of organic pathology? Heart murmur Lung sounds Organomegaly Swollen abdomen Lymphadenopathy Ambiguous genitalia Nervous system deficit Are there signs of MNDs? Pallor Rickets stigmata Stomatitis/cheilosis Oedema Dermatitis Skin/nails abnormalities
Developmental assessment	What age were the milestones reached? Does he/she have motor impairment? Is the language appropriate for age? Does he/she have any learning and/or memory difficulties? Does he/she have any behavioural problems? Does he/she have any social and/or emotional problems?

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
