# Peer review of "Failure to Thrive in the Outpatient Clinic: A New Insight"

_nutrients, 2020, doi:10.3390/nu12082202_

Round 1
Reviewer 1 Report
Synopsis
This is a review of the disorder known as Failure to Thrive (FTT). The authors provide both the classic and a recommended new definition that incorporates malnutrition related to macronutrient and micronutrient intake. The authors report the epidemiology of FTT, which they recommend calling Faltering Growth (FG) and they discuss the etiology of FG. They discuss a variety of factors related to the development of FG. They present an organized approach to evaluation of a child with FG and they recommend a broad approach to treatment of FG.
Critique
This is a very well written review of an important pediatric disorder. This will help providers recognize failure to thrive or faltering growth in their patients and offer a thoughtful approach to diagnosis and treatment.
Author Response
Dear Reviewer,
thank you for your feedback. We are getting the English checked.
Yours sincerely
Reviewer 2 Report
Title: Failure to thrive in the outpatient clinic – a new insight
Manuscript ID: Nutrients-869037
Description:
Failure to thrive (FTT) is a term used to describe inadequate growth or the inability to maintain growth, usually in early childhood. It is a sign of undernutrition, and because many biologic, psychosocial, and environmental processes can lead to undernutrition, FTT should never be a diagnosis unto itself. A careful history and physical examination can identify most causes of FTT, thereby avoiding protracted or costly evaluations. This review offers a wider conception of FTT as an unsatisfactory nutritional state correlated to poor health, looking beyond the “classical” anthropometric criteria for its definition and treatment. This is a well written review, this review required minor modification. Reviewer has some suggestion to improve the quality of this review.
Comments
- Author can include more information in the new insight there is only minimal information was provided by the authors in the current manuscript, author can refer more recent paper to get more input.
- In epidemiology section author can include a table that represent the world wide case reports. This will increase the scope of this review.
- In treatment section author can include more information and reviewer suggested that include a table that represent past and present treatments carried worldwide.
Author Response
Dear Reviewer,
thank you for your suggestion. Here we summarize the main changes made:
- Author can include more information in the new insight there is only minimal information was provided by the authors in the current manuscript, author can refer more recent paper to get more input.
We extended the paragraph "A new insight", including some more references. Being this approach a new proposal, further studies and observation are needed to validate and standardize it. - In epidemiology section author can include a table that represent the world wide case reports. This will increase the scope of this review.
Unfortunately, data on worldwide prevalence of faltering growth in the outpatients setting are not available. Nonetheless, we added some link with updated WHO data on prevalence of MNDs and child growth and malnutrition. - In treatment section author can include more information and reviewer suggested that include a table that represent past and present treatments carried worldwide.
As suggested, we added a figure summarizing past and suggested new approach to faltering growth.
Please don't hexitate if you have any further suggestion.
Yours sincerely,
